# Assessment of Novel Genetic Diversity Induced by Mutagenesis and Estimation of Genetic Parameters in Sesame M4 Mutant Lines

Mohamed Kouighat [1,2], Hafida Hanine [2], Oumaima Chetto [1], Samir Fakhour [3], Mohamed El Fechtali [1] and Abdelghani Nabloussi [1,*]

[1] Research Unit of Plant Breeding and Plant Genetic Resources Conservation, Regional Agricultural Research Center of Meknes, National Institute of Agricultural Research, Avenue Ennasr, P.O. Box 415, Rabat 10090, Morocco

[2] Laboratory of Bioprocess and Biointerfaces, Department of Biology, Faculty of Sciences and Technics, University Moulay Slimane, P.O. Box 523, Beni-Mellal 23000, Morocco

[3] Research Unit of Irrigated Production Systems, Regional Agricultural Research Center of Tadla, National Institute of Agricultural Research, Avenue Ennasr, P.O. Box 415, Rabat 10090, Morocco

* Correspondence: abdelghani.nabloussi@inra.ma

**Abstract:** Sesame (*Sesamum indicum* L.) is an ancient oilseed, aromatic, and medicinal crop widely used for its high-quality oil and seeds. The available genetic diversity in Morocco is too limited; thus, a mutagenesis breeding program was adopted. This study was carried out to evaluate the novel variability induced and observed in 11 M4 mutant lines and to estimate some valuable genetic parameters. The experiment was conducted in two different environments using a randomized complete block design with three replications. Phenological, morphological, and agronomic traits were recorded. To estimate the effect of genotype, environment, and their interaction, ANOVA and planned contrast analyses were performed. To examine relatedness among genotypes, cluster analysis was performed. Significant differences among mutants and between parent cultivars and their respective mutant lines were observed. Genetic parameters such as genotypic (GCV) and phenotypic (PCV) coefficients of variation, broad-sense heritability ($H^2$ b.s), genetic advance (GA), and genetic advance over the mean (GAM) were high in most traits. Highly productive mutants, 'US2-1' and 'US1-2', were observed, exhibiting the highest number of capsules per plant and seed yield ever reported. Additionally, there are other promising mutants with early flowering, early maturity, and a reduced height of the first capsule. This suggests that mutagenesis can be successfully applied to develop high-yielding sesame varieties along with other improved phenological and agromorphological traits. All these mutant lines can be used as promising germplasm to develop competitive sesame cultivars to meet the increasing demand for sesame oil and seeds in the actual context of climate change.

**Keywords:** sesame; mutagenesis; agronomic trait; heritability; genetic advance; early flowering; high seed yield

## 1. Introduction

Sesame (*Sesamum indicum* L.) is an ancient oilseed, aromatic, and medicinal crop widely cultivated for its high-quality oil and seeds. It is an herbaceous tropical and subtropical plant; however, it is also cultivated in arid and semi-arid regions. Sesame seeds are highly nutritious (oil 40–62%, protein 15–25%, carbohydrates 13.4–25.0%, and digestible fiber 9.8%), and the oil is rich in natural antioxidants such as sesamin and sesamol [1]. Sesame seeds are also rich in vitamins, namely, vitamin E, vitamin C, vitamin A, thiamine (B1), riboflavin (B2), niacin (B3), pyridoxine (B6) and folate (B9), minerals (calcium, potassium, pantothenic acid, phosphorus, iron, magnesium, zinc), and tocopherols [1,2]. The seeds and oil are, thus, used

for various purposes such as nutrition, medicine, and industry [2]. Recently, due to human population growth along with consumer awareness of high-quality nutrition, the demand and commercial value of sesame oil and seeds has increased [3]. In fact, over the world, sesame is grown on a global area of about 12.82 million hectares (ha) and Sudan, India, and Myanmar lead in terms of the harvested area, with 5,173,521, 1,520,000 and 1,500,000 ha, respectively [4]. However, the global acreage of sesame in Morocco started to scale back over the last decade. In fact, between 2007 and 2021, the area harvested has decreased by more than 65%, from 2512 to 849 ha. This decline is accompanied by a strong yield fluctuation since 2012 [4]. Sesame is a minor oilseed crop in Morocco, mainly cultivated as a catch crop in the Tadla area (Beni Mellal-Khenifra region ensures more than 90% of the national production). Thus, we must continuously develop new varieties that can meet the demand of various agroecological regions, as well as the changing consumer habits. All the more, the genetic diversity of sesame, in Morocco, is very limited [5,6], and the only cultivar with wild properties, namely, long cycle, dehiscent capsules, indeterminate growth, etc., has been over cultivated by farmers [7]. Over the world, a wide range of genetic diversity has been revealed in most of the producing countries of this crop, however, that existing among the African cultivars is the lowest [8–11].

The success of any varietal breeding program depends on the extent of genetic variability present in the parent material. Nevertheless, without the ability to draw from a diverse genetic pool, the induction of new variations offers an alternative solution via hybridization and mutagenesis [12]. Mutagenesis has been used successfully in oilseed crops such as sesame [13–18]. Therefore, selection by mutagenesis using EMS (ethyl methane sulfonate) was chosen to expand the genetic variability in Moroccan sesame. As a result, many mutants were obtained and phenotypically characterized and monitored during $M_1$, $M_2$, and $M_3$ generations. From those, 11 $M_4$ mutants were selected based on phenotypic criteria, such as seed coat color, phenology, and plant architecture. However, further and specific genetic divergence analysis is needed to screen and identify promising sesame mutants for the future development of high seed-yielding varieties.

Since yield and its components are quantitative and polygenic traits, their phenotypic variation is attributable to genetic variability, environmental influence, and genotype–environment interactions [19]. Therefore, information on the extent and inheritance of induced variability is mandatory for breeding improved varieties. According to Wacal et al. [20], estimation of trait variance and heritability is crucial in breeding programs, as these parameters determine the inherited component of quantitative traits. As a result, genetic characterization and estimation of genetic parameters must involve diverse genetic material evaluated in various environments. Nevertheless, estimating broad-sense heritability alone is not reliable for breeding, as its value can be affected by environmental conditions and plant materials [21]. Patil and Lokesha [22] reported that high broad-sense heritability, coupled with high genetic advance, is more desirable and effective in predicting trait selection than heritability alone.

Although genetic variability, heritability, and genetic advancement for agronomic traits have been reported in various sesame germplasms, such studies are quite rare in mutation-induced variability in sesame [15,17,23–25]. It is interesting to understand the impact of induced mutations on genetic variability, heritability, and the genetic progress of sesame agromorphological traits. Under this genesis, the present study was undertaken to assess the extent of genetic variability for some specific traits and genetic parameters of the 11 $M_4$ sesame mutant lines.

## 2. Materials and Methods

### 2.1. Plant Material

The plant material used in the present study consisted of 13 sesame genotypes, including 11 $M_4$ mutants, and their two wild-type parents (check cultivars). Table 1 illustrates the main phenotypic traits of the same.

**Table 1.** Main phenotypic characteristics of the 13 sesame genotypes studied.

| Lines | Origin | Characteristics |
|-------|--------|-----------------|
| 'ML13' | Local cultivar from Morocco | Parent (check cultivar), beige seeds |
| 'ML2-10' | | Mutant, brown seeds, high branching, small capsules |
| 'ML2-37' | Developed by EMS-mutagenesis from ML13 | Mutant, brown seeds, high branching, low number of capsules per plant |
| 'ML2-5' | | Mutant, brown seeds, tall plant |
| 'ML2-68' | | Mutant, grey seeds, late maturity |
| 'ML2-72' | | Mutant, brown seeds, late flowering |
| 'US06' | Accession from Mexico | Parent (check cultivar), white seeds |
| 'US1-2' | | Mutant, white seeds, high number of capsules per plant |
| 'US1-3' | | Mutant, white seeds, early maturity |
| 'US1-DL' | Developed by mutagenesis from US06 | Mutant, white seeds, late flowering, high branching |
| 'US2-1' | | Mutant, white seeds, early maturity, high number of capsules per plant |
| 'US2-6' | | Mutant, black seeds, early flowering and maturity |
| 'US2-7' | | Mutant, white seeds, low branching |

The 11 mutant lines were derived from a mutagenesis breeding program that was launched in 2019 by our team to expand the narrow existing genetic variability [26]. To do that, healthy and mature seeds of the two sesame genotypes (ML13 and US06) were treated with two concentrations of EMS (ethyl methane sulfonate), 0.5% and 1% for five hours. The mutant plants thus observed were monitored and evaluated during three generations. Seeds of the mutant lines are propagated in isolation to keep the authenticity of the genetic material as cross pollination, even with a low rate, can occur in sesame. In fact, the selected mutant plants are bagged before flowering to ensure self-fertilization by avoiding any contamination by allopollen. Finally, 11 $M_4$ mutants were selected based on some interesting traits such as seed coat color, early/late flowering and maturity, plant branching, and plant capsules abundance.

### 2.2. Field Experiment

The study was carried out in two different environments. The first one is the experimental station of Ain Taoujdate (E1), belonging to the National Institute of Agronomic Research (INRA), Regional Center of Meknes, Morocco. This area is located in the province of El Hajeb—the region of Fez-Meknes, Morocco (2°13′00″ N 6°30′00″ W), at an altitude of 550 m and a continental climate with 470 mm of precipitation and maximum temperature in August (45 °C) and minimum in January (2.8 °C). The soil is clayey, calcareous brown, and alluvial. The second environment is the experimental station of Afourare (E2), located in the province of Beni-Mellal, Morocco (33°55′59″ N 5°16′28″ W), at an altitude of 446 m, characterized by a chronic luvisol soil, a semi-arid climate with 400 mm of precipitation, and a maximum temperature 46 °C in August and minimum 0 °C in January. During this study, the climatic data (Table 2) were obtained from automated weather stations placed at the two experimental sites.

To homogenize germination, seeds of each genotype (harvested in 2020) were sown on 22 April 2021 in plastic cells. After three weeks of emergence, i.e., at the second pair of leaves, the plants were transplanted into both environments (E1 and E2). The experiment was set up using a randomized complete block design (RCBD) with three replicates per location. Each genotype was sown in plots of two rows of two meters long each, with 60 cm row spacing and 15 cm between plants. All cultural practices were applied manually, as recommended by Langham [27] for sesame production at each location. Afterward, the 13 sesame genotypes were evaluated in both E1 and E2 environments from April to October 2021.

**Table 2.** Climatological data of the two environments during the study period.

| | Afourare (E2) | | | | Taoujdate (E1) | | | |
| | Temperature [°C] | | | Precipitation [mm] | Temperature [°C] | | | Precipitation [mm] |
| Date | Min | Max | Mean | Sum | Min | Max | Mean | Sum |
| --- | --- | --- | --- | --- | --- | --- | --- | --- |
| 2021-04 | 04.64 | 25.89 | 19.80 | 55.80 | 06.64 | 22.2 | 16.19 | 72.20 |
| 2021-05 | 03.65 | 35.64 | 28.75 | 36.80 | 05.93 | 23.6 | 20.32 | 23.60 |
| 2021-06 | 07.40 | 36.28 | 30.50 | 02.40 | 10.42 | 31.21 | 26.09 | 1 |
| 2021-07 | 11.23 | 45.44 | 35.73 | 0 | 13.37 | 42.97 | 33.27 | 0 |
| 2021-08 | 09.73 | 46.00 | 44.20 | 0 | 12.18 | 43.20 | 41.65 | 0.20 |
| 2021-09 | 11.04 | 42.41 | 41.36 | 09.60 | 13.91 | 31.6 | 29.74 | 01.60 |
| 2021-10 | 10.12 | 37.37 | 31.12 | 03.50 | 13.42 | 28.14 | 22.89 | 04.80 |

### 2.3. Data Collecting and Parametrs Studied

At harvest, a sample of ten plants was randomly selected from each plot, in both environments, to assess phenological, morphological, and agronomic traits. The morphological traits were plant height (PH, cm), fruiting zone length (FZL, cm), height of the first capsule (HFC, cm), and the number of primary branches/plant (NBP). The phenological parameters consisted of the number of days from sowing to flowering (NDF, d) and maturity (NDM, d). Finally, the agronomic traits were the number of capsules/plant (NCP), number of seeds/capsule (NSC), weight of 1000 seeds (TWS, g), and seed yield (kg·ha$^{-1}$).

Then, various genetic parameters were estimated. Genotypic (Vg) and phenotypic (Vp) variances were calculated according to Singh and Chaudhry [28], using the formulas (i) and (ii). Genotypic (GCV) and phenotypic (PCV) coefficients of variation, broad-sense heritability (H$^2$ b.s, %), genetic advance (GA), and genetic advance over the mean (GAM, %) were estimated according to Deshmukh et al. [29], using the expressions (iii), (iv), (v), (vi), and (vii). The phenotypic and genotypic coefficients of variation were considered high (>20%), moderate (10–20%), or low (<10%) according to the scale described by Deshmukh et al. [29]. Broad sense heritability values were also grouped as high (>50%), moderate (20–50%), or low (< 0%), as proposed by Deshmukh et al. [29].

i.      Genotypic variance $Vg = (MSg - MSe)/r$, where $MSg$ is the genotype mean square, $MSe$ is the mean square of experimental error, and $r$ is the number of replicates.

ii.     Phenotypic variance: $Vp = Vg + MSe$

iii.    Phenotypic coefficient of variation: PCV $= \sqrt{(Vp/(\bar{x}))} \times 100$

iv.     Genotypic coefficient of variation: GCV $= \sqrt{(Vg/(\bar{x}))} \times 100$

v.      Broad sense heritability: $H2\ b.s = (Vg \times 100)/Vp$

vi.     Genetic advance: GA $= k \times H2\ b.s \times \sqrt{Vp}$

vii.    Genetic advance over mean: $GAM = \frac{GA \times 100}{\bar{x}}$, where $\bar{x}$ is the grand mean and k = 2.06 is the intensity of selection at 5% [30].

### 2.4. Statistical Analyses

All results were statistically analyzed using SPSS computer software (SPSS version 21, Chicago, IL, USA). The data obtained from the two environments were analyzed independently and then combined. Both the environment and genotype factors are considered fixed. An analysis of variance (ANOVA) was performed to reveal differences among genotypes under different environments. To compare all the mutant lines and their wild-type parents, Duncan's new multiple plaque test (DMRT) was performed. Furthermore, planned contrasts (one-way ANOVA) were carried out to test the significance of differences observed between checks cultivars (ML13 and US06) and their respective mutants, as well as between the Moroccan ML genotypes and the foreign US genotypes. The variance results were used to estimate the coefficients of phenotypic, genotypic, environmental, heritability variability, genotypic progress, and mean genetic progress. Additionally, Pearson's correlation

coefficients were computed to examine the associations among the parameters studied. Finally, a dendrogram was constructed by ascending hierarchical clustering to examine agromorphological diversity and relatedness between the genotypes studied.

## 3. Results and Discussion

### 3.1. Effects of Genotype, Environment and Their Interaction

The analysis of individual and combined variances (Table 3) showed a highly significant effect of genotype ($p < 0.001$) on all the studied traits. Additionally, the effect of the environment was very significant on all the traits, except for 1000-seed weight. The combined ANOVA also revealed a significant effect of genotype × environment (G × E) interaction on all the traits studied, except for the number of seeds per capsule.

Mean squares from analysis of variance revealed significant variation among mutants and both check cultivars. Moreover, planned contrast analysis by ANOVA showed significant differences between the US06 parent and the US mutants for each of the ten traits measured/calculated (US06 parent vs. US mutants) (Table 4). Additionally, the ML mutants differed significantly from their check parent ML13 (ML13 parent vs. ML mutants) for all traits except NDF. Likewise, there were significant differences between the ML genotypes group and the US genotypes set for all parameters. Overall, wide genetic variability was induced by the mutagenesis for different phenological, morphological, and agronomic traits in the sesame germplasm. Previous studies have also reported the possibility of improving the genetic variability of sesame by mutagenesis [15,17,18,24,25,31–39].

**Table 3.** Results of individual and combined analysis of variance (mean squares) for ten traits in 13 sesame genotypes evaluated in two environments.

| Source of Variation | DF | Number of Days to Flowering (d) | Number of Days to Maturity (d) | Height of the First Capsule (cm) | Plant Height (cm) | Fruiting Zone Length (cm) | Number of Branches per Plant | Number of Capsules per Plant | Number of Seeds per Capsule | 1000-Seed Weight (g) | Seed Yield (kg·ha⁻¹) |
|---|---|---|---|---|---|---|---|---|---|---|---|
| | | | | | Afourare (E1) | | | | | | |
| Block | 2 | 30.11 * | 18.02 | 14.64 | 40.38 * | 20.84 | 0.10 | 69.46 * | 4.38 | 0.02 | 9.25 |
| Genotype | 12 | 2609.70 *** | 1854.62 *** | 2656.01 *** | 2282.14 *** | 2317.41 *** | 228.64 *** | 62,916.86 *** | 189.58 *** | 0.21 *** | 1998.83 *** |
| Error | 24 | 29.86 | 30.41 | 39.32 | 31.41 | 41.32 | 2.08 | 234.77 | 8.10 | 0.01 | 19.42 |
| | | | | | Taoujdate (E2) | | | | | | |
| Block | 2 | 39.93 * | 14.18 * | 11.47 | 61.34 * | 23.27 | 1.45 | 67.41 * | 5.671 | 0.03 | 11.13 * |
| Genotype | 12 | 191.70 *** | 1162.15 *** | 2952.30 *** | 5318.92 *** | 7854.32 *** | 146.87 *** | 24,026.92 *** | 258.41 *** | 0.29 *** | 853.95 *** |
| Error | 24 | 23.56 | 100.63 | 47.09 | 18.28 | 77.53 | 5.40 | 349.30 | 8.42 | 0.09 | 33.62 |
| | | | | | Combined | | | | | | |
| Environment (E) | 1 | 490.43 *** | 10,674.17 *** | 347.343 *** | 2829.14 *** | 20,129.17 ** | 74.818 *** | 49,025.88 *** | 64.77 *** | 0.09 | 20.12 *** |
| Block | 4 | 35.02 | 16.10 | 13.05 | 50.86 | 22.05 | 0.77 | 68.43 | 5.026 | 0.025 | 19.19 |
| Genotype (G) | 12 | 267.5 *** | 1463.3 *** | 6064.5 *** | 2312.8 *** | 3175.6 *** | 197.7 *** | 24,878.6 *** | 233.4 *** | 0.282 *** | 1330.6 *** |
| G × E | 12 | 48.3 * | 422.3 *** | 2063.60 *** | 2063.60 *** | 2043.10 *** | 30.90 *** | 17,198.10 *** | 11.90 | 0.062 *** | 18.8 *** |
| Error | 48 | 26.71 | 65.52 | 43.205 | 24.845 | 59.425 | 3.74 | 292.035 | 8.26 | 0.05 | 26.52 |

DF: Degree of freedom. *, ** and ***: significant differences at 5%, 1%, and 0.1% probability levels, respectively.

**Table 4.** Results of analysis of variance of 13 sesame genotypes (11 mutant lines and their two parents) evaluated for various traits (mean square, the value of contrast, and level of difference significance).

| Source of Variation | DF | Number of Days to Flowering (d) | Number of Days to Maturity (d) | Height of the First Capsule (cm) | Plant Height (cm) | Fruiting Zone Length (cm) | Number of Branches per Plant | Number of Capsules per Plant | Number of Seeds per Capsule | 1000-Seed Weight (g) | Seed Yield (kg·ha⁻¹) |
|---|---|---|---|---|---|---|---|---|---|---|---|
| | | | | | Mean square | | | | | | |
| Genotype | 12 | 2609.70 *** | 1854.62 *** | 2656.01 *** | 2282.14 *** | 2317.41 *** | 228.64 *** | 62,916.86 *** | 189.58 *** | 0.21 *** | 198.83 *** |
| | | | | | Contrast value | | | | | | |
| ML13 parent vs. ML mutants | 1 | 0.90 | 10.30 * | 46.02 ** | 23.20 ** | 77.70 ** | 10.54 ** | 302.26 *** | 76.80 ** | 1.50 *** | 1.23 * |
| US06 parent vs. US mutants | 1 | 27.10 * | 34.40 * | 33.30 ** | 9.22 * | 27.70 * | 18.40 *** | 130.40 ** | 24.90 * | 0.95 *** | 2.40 ** |
| US genotypes vs. ML genotypes | 1 | 73.80 ** | 118.60 *** | 161.02 *** | 17.47 ** | 125.70 *** | 41.45 *** | 306.66 *** | 34.10 * | 0.59 *** | 3.43 *** |

DF is the degree of freedom. *, ** and ***: significant differences at 5%, 1%, and 0.1% probability levels, respectively.

*3.2. Mean Performances of Genotypes*

3.2.1. Number of Days to Flowering

The results revealed the presence of a wide genetic variability for number of days to flowering (NDF) (Tables 3 and 5) among the genotypes studied, mainly between the ML genotypes group and US genotypes group ($p < 0.01$) and between the check cultivar (US06) and its derived US mutants ($p < 0.05$) (Table 4). The GCV value (9.68) was slightly lower than the PCV value (12.50), indicating minor environmental effects on NDF gene expression. This trait had a high $H^2$ b.s (60.04%), coupled with a moderate GA (13.05) (Table 5). The average values of NDF vary between 55.80 and 72.5 days (d) with an average of 65.46 d. This wide range of variation makes it possible to select and develop new early flowering varieties that are more adapted to the local environment and climate change. From a breeding perspective in sesame, early flowering is desirable because it would result in more capsules and, consequently, more seed yield [40]. The two mutants 'ML2-68' and 'US1-DL' were the latest in terms of flowering with an average NDF of 72.5 d (Figure 1), whereas the 'US1-3' mutant was the earliest in flowering with an NDF of 55.80 d, followed by the 'US2-6' mutant (56.15 d). These two mutants bloomed about 12 days before the Moroccan cultivar (ML13), which began to flower at 67.93 d. Regarding flowering phenology, our mutants proved to be earlier than those reported by Laghari et al. [25] having a NDF of 57.33 d in sesame mutants with a $H^2$ b.s of 97.32% and a GA of 8.41. However, they are slightly later than the mutants found by Aliyu et al. [36] that flowered at 42.25 days after sowing, with an $H^2$ b.s of 89.73%.

**Table 5.** Estimation of genetic parameters (genetic variance, phenotypic variance, heritability, and genetic advance) in 13 sesame genotypes for various phenological and agromorphological traits.

| Characters | Range | Mean | Component of Variance | | | Genetic Variability | | | Genetic Advance | |
| | | | Vg | Vp | Ve | PCV | GCV | H² b.s% | GA | GAM% |
|---|---|---|---|---|---|---|---|---|---|---|
| NDF | 55.8–72.5 | 65.46 | 40.13 | 66.84 | 26.71 | 12.50 | 9.68 | 60.04 | 13.05 | 19.95 |
| NDM | 129.08–166.63 | 145.81 | 232.97 | 298.49 | 65.52 | 11.85 | 10.47 | 78.05 | 31.44 | 21.56 |
| HFC | 19.96–77.70 | 47.21 | 1003.56 | 1046.76 | 43.21 | 68.53 | 67.10 | 95.87 | 65.26 | 138.23 |
| PH | 117.6–180.67 | 154.43 | 381.34 | 406.18 | 24.85 | 13.05 | 12.65 | 93.88 | 40.23 | 26.05 |
| FZL | 82.92–128.81 | 110.05 | 519.37 | 578.80 | 59.43 | 21.70 | 20.55 | 89.73 | 46.95 | 42.34 |
| NBP | 2.5–15.8 | 7.94 | 32.34 | 36.08 | 3.74 | 74.62 | 70.65 | 89.63 | 11.72 | 145.53 |
| NCP | 169.10–341 | 248.08 | 4097.77 | 4389.80 | 292.04 | 27.20 | 26.28 | 93.35 | 131.87 | 54.13 |
| NSC | 58.30–67.60 | 62.96 | 37.53 | 45.79 | 8.26 | 10.72 | 9.70 | 81.96 | 12.62 | 19.99 |
| TSW | 2.99–3.60 | 3.25 | 0.04 | 0.09 | 0.05 | 9.19 | 6.07 | 43.61 | 0.41 | 12.50 |
| Yield | 516–1691 | 1052 | 217.35 | 243.87 | 26.52 | 196.19 | 185.21 | 89.13 | 30.37 | 381.54 |

Genetic variance (Vg), phenotypic variance (Vp), environmental variance (Ve), phenotypic (PCV) and genotypic (GCV) coefficients of variation, broad-sense heritability ($H^2$ b.s, %), genetic advance (GA), genetic advance over the mean (GAM, %), number of days to flowering (NDF, d), number of days to maturity (NDM, d), height of the first capsule (HFC, cm), plant height (PH, cm), fruiting zone length (FZL, cm), number of primary branches per plant (NBP), number of capsules per plant (NCP), number of seeds per capsule (NSC), 1000-seed weight (TSW, g), and seed yield (Yield, kg·ha$^{-1}$).

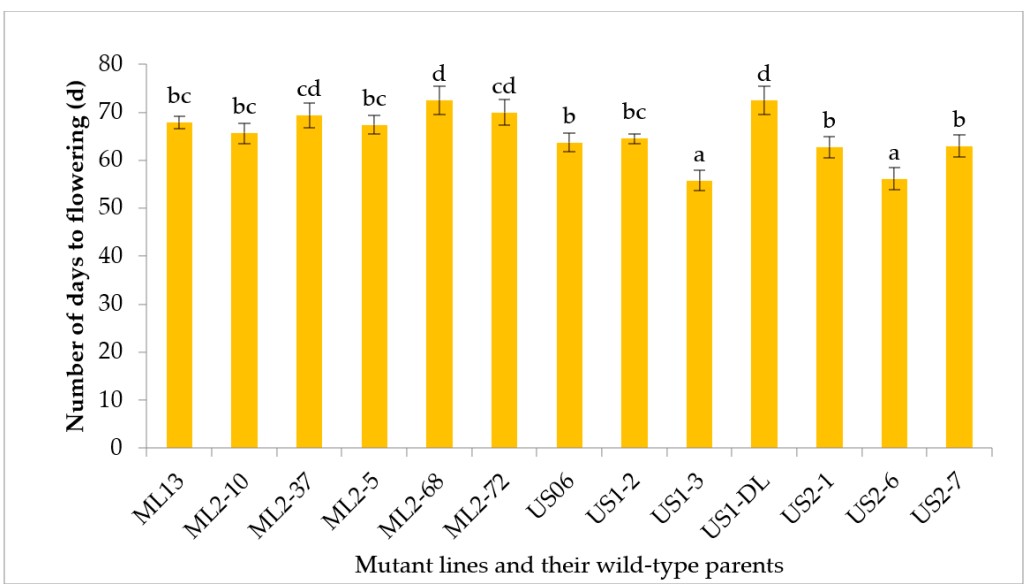

**Figure 1.** Number of days to flowering (NDF) in 11 sesame mutant lines and their two wildtypes (US06 and ML13). The superscript letters represent the groups according to Duncan's test.

### 3.2.2. Number of Days to Maturity

An appreciable genetic (232.97) and phenotypic (298.49) variability was found in the number of days to maturity (NDM) trait (Table 5). The contrast analysis values display that the overall variance was mostly obtained between US genotypes and ML genotypes ($p < 0.001$) followed by the check parent 'US06' and the mutants US ($p < 0.05$) and then between the ML check parent and the mutants ML ($p < 0.05$) (Table 4). The average duration of plant maturity ranged from 129.08 to 166.63 d with an overall average of 145.81 d (Table 5). The mutants 'ML2-68' and 'ML2-72' matured late after 166.63 and 161.60 d, respectively, while the two mutants 'US2-6' and 'US2-1' matured early at 129.30 and 130.20 d, respectively (Figure 2). These two mutants were earlier than the check cultivar 'US06' by seven days and the Moroccan check cultivar (ML13) by 24 days. Moderate values of PCV (11.85), slightly higher than GCV (10.47), were observed for this trait, implying that the environment has a low effect on NDM's gene expression, which allows easy selection. A high $H^2$ b.s (78.05%), coupled with a high GA (31.44), offers the advantage of using 'US2-6' and 'US2-1' mutants in a breeding program for early maturing varieties. The other two mutants, 'ML2-68' and 'ML2-72', will be useful for the development of late varieties. The aforementioned mutants mature later than those described by Bhuiyan et al. [38] (NDM of 83 d after sowing, $H^2$ b.s of 62.51% coupled with a low GA of 5.89). Indeed, the late maturity in sesame plants was positively correlated with high NCP and seed yield [41]. On the other hand, our mutant 'US2-6' (NDM of 129.30 d, $H^2$ b.s 78.05% coupled with a high GA, 31.44) was slightly earlier than that found by Laghari et al. [25] (NDM of 131.30 d, $H^2$ b.s 99.27%, and low GA, 12.30). These findings revealed that the phenology of the sesame plant can be effectively modified by mutagenesis. Thus, early and late maturity is important for breeding programs trying to adapt sesame cultivars to various and particular ecological regions.

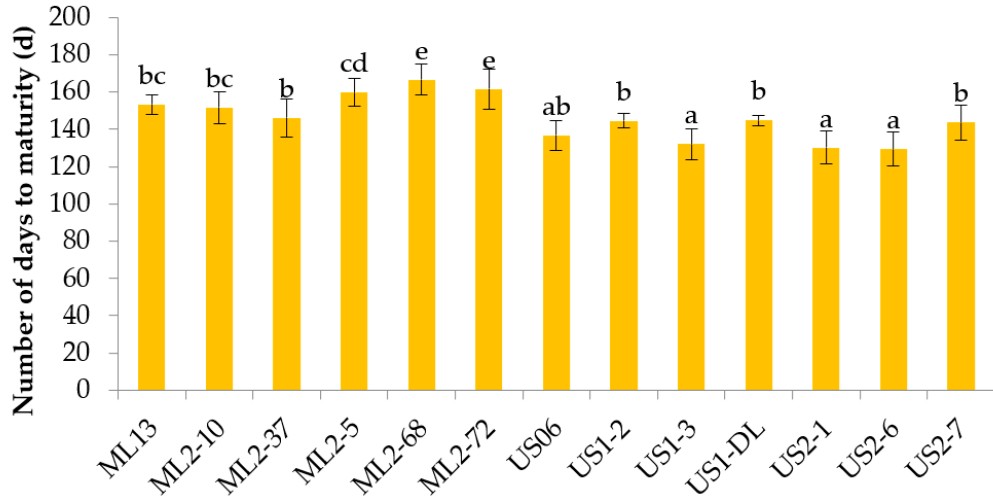

**Figure 2.** Number of days to maturity (NDM) in 11 sesame mutant lines and their two wildtypes (US06 and ML13). The superscript letters represent the groups according to Duncan's test.

### 3.2.3. Height of the first Capsule

Very high genotypic (1003.56) and phenotypic (1046.76) variances were observed for height of the first capsule (HFC) (Table 5). The largest differences ($p < 0.001$) existed between the US genotypes and the ML genotypes, followed by those between the ML parent and the ML mutants ($p < 0.01$) and between the US parent and the US mutants ($p < 0.05$) (Table 4). This emphasizes the presence of a large genetic variability induced by mutagenesis for this trait. The very close values of PCV (68.53) and GCV (67.10) indicate minor environmental effects on HFC gene expression (Table 5). Additionally, a high $H^2$ b.s (95.87%), coupled with a high GA (65.26), were found for this trait. Therefore, phenotypic selection for HFC will be effective. The mean HFC value was 47.21 cm, varying from 19.96 to 77.70 cm (Table 5). Mutant 'US2-6' showed the lowest HFC value of 19.96 cm, followed by the check cultivar 'US06' (25.85 cm), the mutant 'US2-7' (27.45 cm), the mutant 'US1-3' (31.45 cm) and the mutant 'US2-1' mutant (34.95 cm) (Figure 3). However, the highest value (77.70 cm) was recorded in the mutant 'ML2-68'. The Moroccan cultivar 'ML13' recorded an average HFC (50.19 cm). Other studies on sesame have reported the possibility of reducing the HFC to 42.50 cm [39] and 30.4 cm [31], which remain much higher than our US mutants. The height of the first capsule is an important characteristic for the mechanical harvesting of sesame. It was reported that a HFC between 15 and 40 cm is desirable for mechanized sesame harvesting [42,43]. Therefore, the genotypes 'US2-6', 'US06', 'US2-7', 'US1-3' and 'US2-1' could be used in a breeding program for mechanized sesame harvesting.

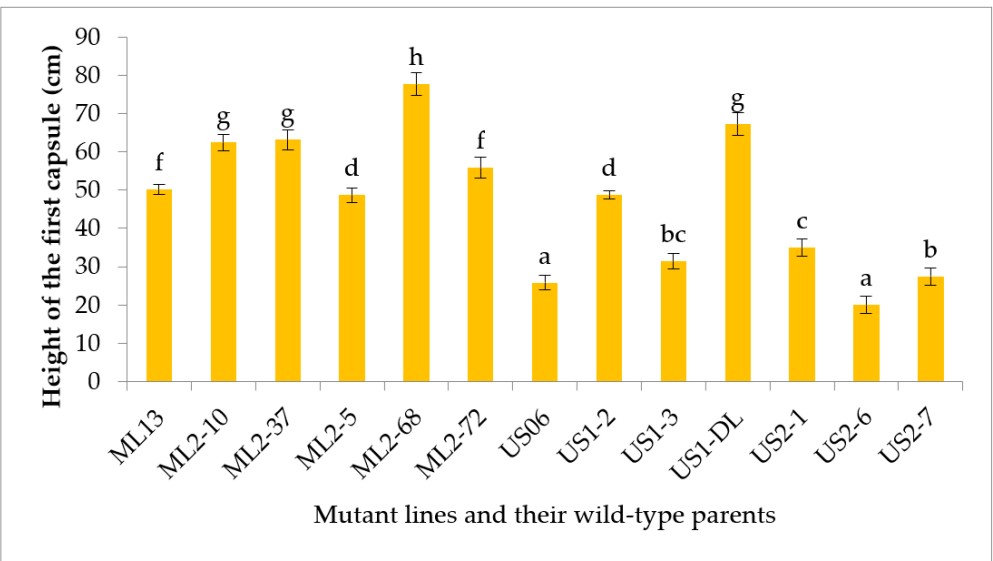

**Figure 3.** Height of the first capsule (HFC) in 11 sesame mutant lines and their two wildtypes (US06 and ML13). The superscript letters represent the groups according to Duncan's test.

### 3.2.4. Plant Height

Plant height (PH) differed significantly among sesame genotypes. Significant differences were observed between parent ML13 and ML mutants ($p < 0.05$), between US genotypes and ML genotypes ($p < 0.05$), and between parent US06 and US mutants ($p < 0.05$) (Table 4). The phenotypic and genotypic variances in PH were 406.18 and 381.34, respectively (Table 5). Moderate GCV and PCV coefficients (12.65 and 13.05, respectively) were recorded for this trait (Table 5). These results inform about the presence of a large genetic variability among mutants and check parents for this trait. Regarding its low variation across different environments, PH presented a very high $H^2$ b.s (93.88%), with an expected GA of 40.23. Plant height ranged from 117.60 to 180.67 cm, with an overall average of 154.43 cm (Table 5). The mutant 'US2-7' was the shortest with a PH of 117.60 cm, followed by the mutant 'US2-6' (137.10 cm). By contrast, the mutant 'ML2-5' was the tallest with a PH of 180.67 cm. The check cultivar ML13 had an average PH of 170.45 cm (Figure 4). This PH range is superior to that of Bhuiyan et al. [38] who reported sesame mutants with a PH between 119 and 136 cm, a high $H^2$ b.s (96.81%), and a moderate GA (14.74). Our PH range is also higher than that of Divya et al. [24] who found sesame mutants with a PH between 47.20 and 132.70 cm and a $H^2$ b.s of 86.20%, coupled with a GA of 32.24. Our results also exceed those reported by Laghari et al. [25], who developed mutant plants with a PH between 128.90 and 161.93 cm, a high $H^2$ b.s of 96.77%, and a GA of 24.62, and those reported by Imran et al. [44] (PH between 25.60 and 141.80 cm, $H^2$ b.s of 92.97% and GA of 42.73). In sesame, plant height is often positively correlated with seed yield due to its indeterminate growth. However, due to strong wind or hurricanes, high plants are susceptible to lodging. To increase lodging resistance and ensure relatively high seed yield, breeders often set plant height reduction as a breeding goal in sesame [45]. The two mutants 'US2-7' and 'US2-6' are the shortest, with a medium seed yield (about 1000 kg·ha$^{-1}$). Therefore, these two mutants could be good genetic sources to develop competitive sesame varieties that can resist lodging.

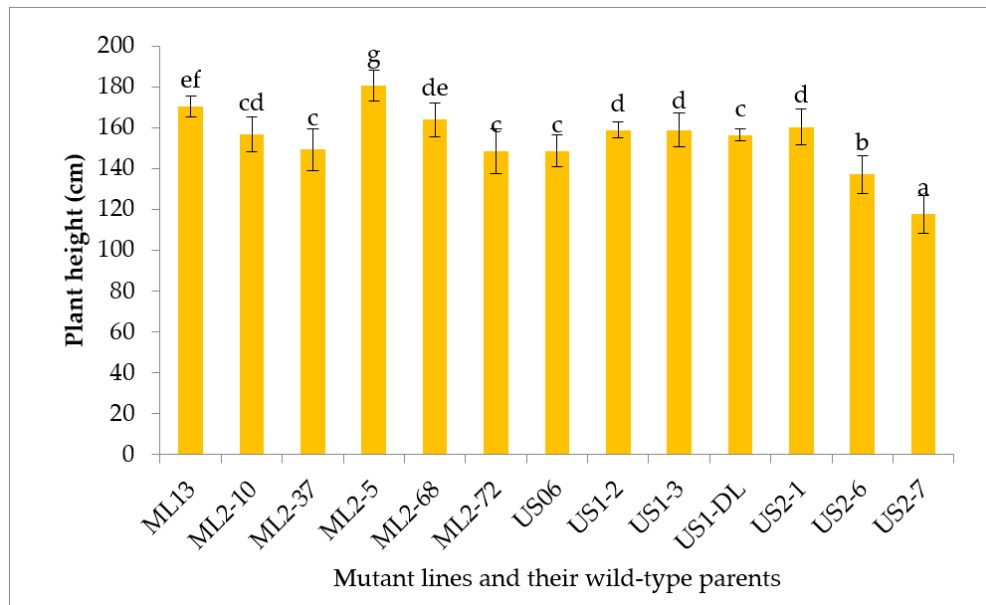

**Figure 4.** Plant height (PH) in 11 sesame mutant lines and their two wildtypes (US06 and ML13). The superscript letters represent the groups according to Duncan's test.

### 3.2.5. Fruiting Zone Length

The results showed a highly significant genotypic variation for fruiting zone length (FZL). The highest differences ($p < 0.001$) were found between the US genotypes and the ML genotypes, followed by the differences between the ML13 parent and the ML mutants ($p < 0.01$) and between the US parent and the US mutants ($p < 0.05$) (Table 4). High PCV and GCV (21.70 and 20.55, respectively), with very high $H^2$ b.s (89.73%), along with high GA (46.95) (Table 5), provide evidence of a broad induced genetic variation, in terms of FZL trait, that could be used in sesame breeding programs. The FZL trait ranged from 82.92 to 128.81 cm, with an overall average of 110.05 cm (Table 5). The shortest FZL values (<100 cm) were found in the mutants 'ML2-68' (82.92 cm), 'ML2-37' (86.06 cm), 'US1-DL' (92.82 cm), and 'ML2-72' (97.67 cm), while the highest (>125 cm) were recorded in the mutants 'ML2-5' (128.81 cm), 'US2-1' (126.66 cm), 'US2-6' (126.02 cm), and the cultivar 'US06' (126.90 cm) (Figure 5). Our mutants' FZL values are greater than those reported by Hoballah [31] (123.4 cm). Despite the importance of the FZL trait in breeding strategies, owing to its direct impact on seed yield in sesame [46], the number of capsules per plant remains the most contributing trait to seed yield. High-yielding mutants, namely, namely, 'US2-1' and 'US1-2' (1691 and 1674 kg·ha$^{-1}$, respectively), have a medium PH (160.40 and 158.90 cm, respectively), recorded a high FZL (126.66 and 122.90 cm, respectively), and a high NCP (341.40 and 331, respectively).

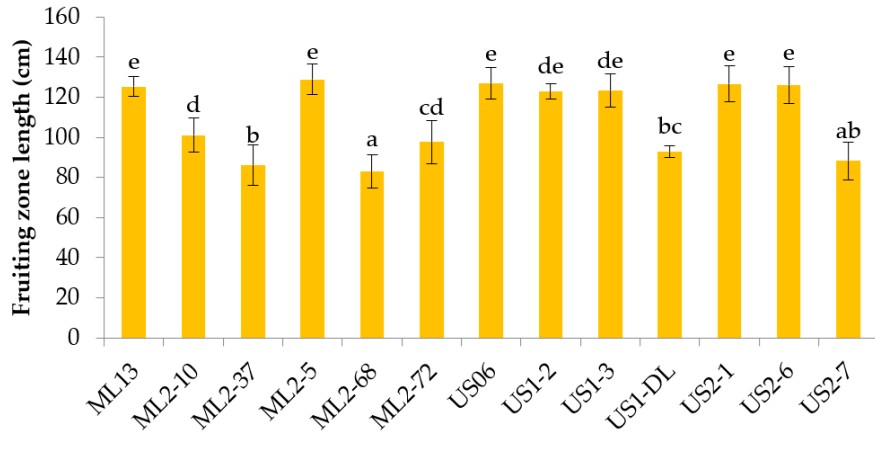

**Figure 5.** Fruiting zone length (FZL) in 11 sesame mutant lines and their two wildtypes (US06 and ML13). The superscript letters represent the groups according to Duncan's test.

### 3.2.6. Number of Primary Branches per Plant

Analysis of variance shows that the number of primary branches per plant (NBP) differed significantly between US genotypes and ML genotypes ($p < 0.001$), parent 'US06' and US mutants $p < 0.001$, and between 'ML13' parent and 'ML' mutants ($p < 0.01$) (Table 4). This trait has a genotypic variance of 32.34 and a phenotypic variance of 36.08, suggesting the presence of valuable genetic polymorphism in NBP (Table 5). The NBP ranged from 2.5 to 15.8, with an overall average of 7.94 branches per plant. The values of PCV (74.62) and GCV (70.65) were close for this trait, indicating its stability in different environments. These results underline the practicality of improving this trait by mutagenesis and, eventually, provide breeders with reliable resources by the end of the following generation. The NBP trait recorded high $H^2$ b.s (89.63%), with a moderate GA (11.72). The least branched genotypes were mutant 'US1-2' (2.5), check cultivar 'US06' (2.8), mutant 'US2-1' (3.2), and mutant 'US2-7' (3.5). In contrast, the mutant 'US1-DL' (15.80), followed by the mutant 'ML2-10' (12.90), and the mutant 'ML2-37' (12.50), account for the abundant branching type (Figure 6). To our knowledge, the mean value of 15.80 observed in the mutant line 'US1-DL' is the highest NBP ever recorded to date. Branched varieties have been recommended to solve the problem of cutting the thick stems of unbranched lines [47]. Branched varieties use the extra space to shade the ground and, therefore, avoid weeds [48].

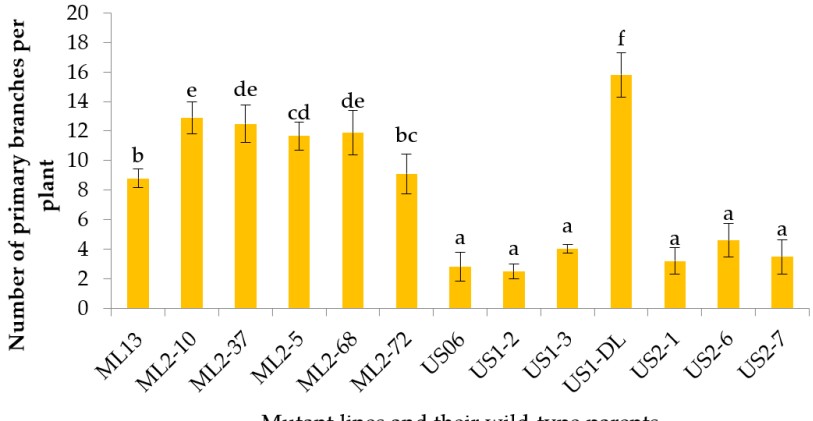

**Figure 6.** Number of primary branches per plant (NBP) in 11 sesame mutant lines and their two wildtypes (US06 and ML13). The superscript letters represent the groups according to Duncan's test.

### 3.2.7. Number of Capsules per Plant

For the number of capsules per plant (NCP) trait, the mutagenesis induced a valuable genetic gap between the ML parent and the ML mutants ($p < 0.001$), on the one hand, and between the US parent and the US mutants ($p < 0.01$), on the other hand (Table 4). This trait recorded high PCV and GCV (27.20 and 26.28, respectively), with high $H^2$ b.s (93.35%), in addition to a high GA (131.87) (Table 5). NCP values ranged from 169.10 to 341.00, with an overall average of 248.08 capsules per plant (Table 5). The mutant 'US1-2' produced the highest NCP (341), closely followed by the mutant 'US2-1' (331.40), while the check cultivar 'US06' produced just 261.20 capsules per plant, which was 23% lower than these mutants 'US1-2' and 'US2-1' (Figure 7). The two mutants, 'US1-2' and 'US2-1', which are the top in terms of NCP, are much more interesting than those of Divya et al. [24] (61 capsules/plant, $H^2$ b.s of 89.2% with a GA of 98.7), those of Imran et al. [44] (85 capsules/plant with an $H^2$ b.s of 98.42 and a GA of 33.95), those of Bhuiyan et al. [38] (87 capsules/plant, $H^2$ b.s of 78.78%, and a GA of 56.2), those of Laghari et al. [25] (166.5 capsules/plant, with an $H^2$ of 93.21%, and GA of 35.73), and those of Begum and Dasgupta [17] (265.77 capsules/plant with an $H^2$ b.s of 87.12% coupled to a GA of 89.78). To the best of our knowledge, this is so far the first report of such high capsules number in the sesame plant. The number of capsules per plant is one of the most important traits in developing the ideal type of high seed-yielding sesame plant [49]. Furthermore, the development of varieties with improved yield contributing traits is the main objective of selecting high seed yield varieties in sesame [50]. Therefore, the two mutants (US1-2 and US2-1) are good genetic resources for developing highly productive sesame varieties.

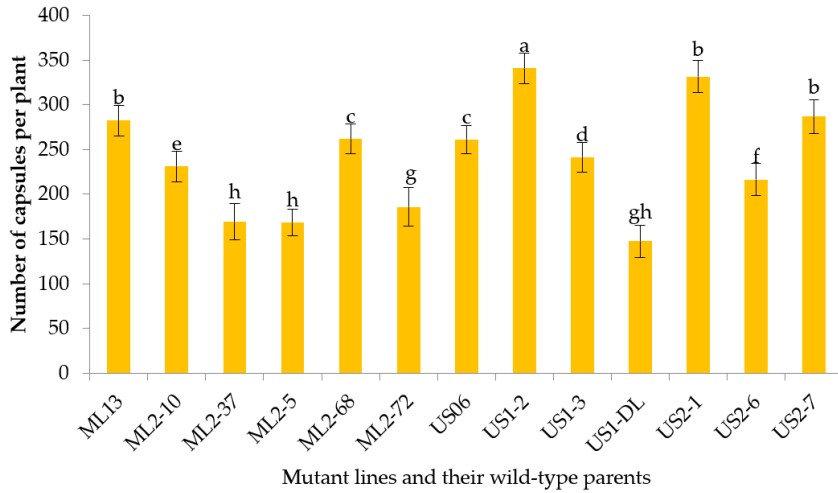

**Figure 7.** Number of capsules per plant (NCP) in 11 sesame mutant lines and their two wildtypes (US06 and ML13). The superscript letters represent the groups according to Duncan's test.

### 3.2.8. Number of Seeds per Capsule

As with the other traits, the number of seeds per capsule (NSC) differed significantly between the ML13 parent and the ML mutants ($p < 0.01$), the US06 parent and the US mutants ($p < 0.05$), and between the ML genotypes and the US genotypes ($p < 0.05$) (Table 4). The PCV and GCV of this trait were moderate at 10.72 and 9.70, respectively (Table 5). This trait recorded a high broad-sense heritability (81.96%) coupled with a moderate genetic advance (12.62). NSC values ranged from 58.30 to 67.60 seeds/capsule with an average value of 62.96 (Table 5). The maximum NSC was observed in the check cultivar 'US06' (67.60), while the minimum one was found in the mutant 'US1-DL' (58.30), followed by the Moroccan cultivar 'ML13' recording a mean value of 59.13 (Figure 8). In contrast to the number of capsules per plant, no mutant exceeded the parent cultivar 'US06' in terms of the number of seeds per capsule. However, previous studies have shown the possibility of improving this trait using mutagenesis [17,38,44].

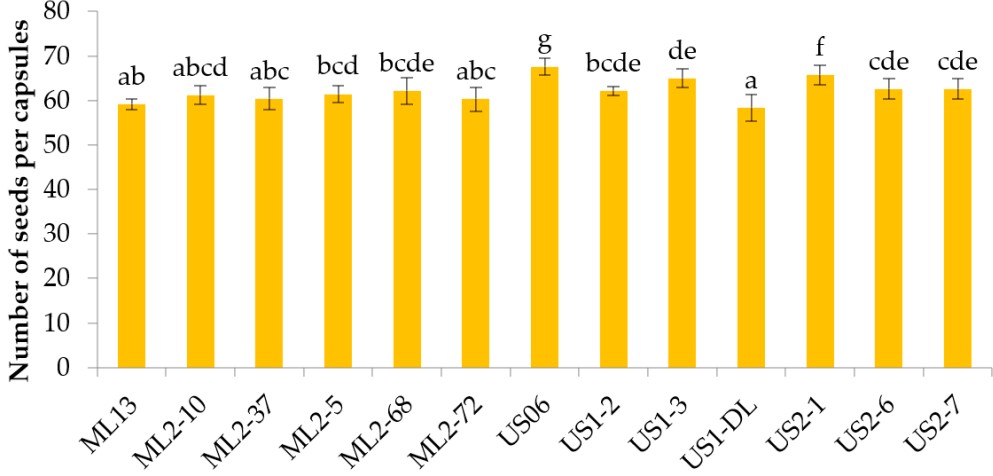

**Figure 8.** Number of seeds per capsules (NSC) in 11 sesame mutant lines and their two wildtypes (US06 and ML13). The superscript letters represent the groups according to Duncan's test.

### 3.2.9. Thousand Seeds Weight

Thousand seeds weight (TSW) varied significantly among sesame genotypes. Highly significant differences ($p < 0.001$) were found between the ML13 parent and the ML mutants, the US06 parent and the US mutants ($p < 0.001$), and between the US genotypes and the ML genotypes ($p < 0.001$) (Table 4). The GCV (6.07) was low and inferior to the PCV (9.19), explaining moderate broad-sense heritability (43.61%) and genetic advance (0.41) for TSW, compared to the rest of the parameters studied. This indicates that TSW is more influenced by environmental conditions than the other parameters. The average TSW was 3.25 g, ranging from 2.99 to 3.60 g (Table 5). The mutant 'US2-6' (2.99 g) and Moroccan cultivar 'ML13' (3.05 g) had the smallest seeds (lightest TSW), while the mutant line 'ML2-72' (3.60 g) recorded the largest seeds (heaviest seeds), being 18% higher than the Moroccan cultivar 'ML13' (Figure 9). The genetic gain registered in our mutants is superior to that of Divya et al. [24] (3.23 g), Begum and Dasgupta [17] (3.39 g), but lower than that of Imran et al. [44] (3.90 g), and Laghari et al. [25] (3.93 g). Our findings indicated a low genetic advance in TSW, which may not be worth selecting promising genotypes for this trait.

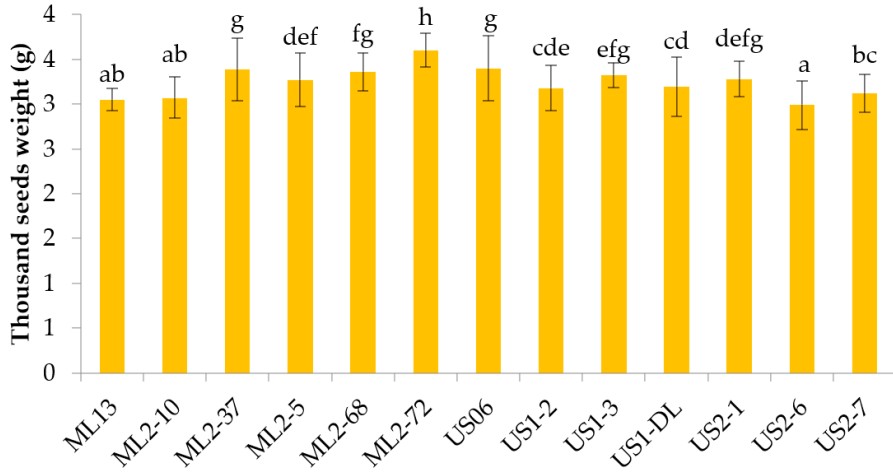

**Figure 9.** Thousand seeds weight (TSW) in 11 sesame mutant lines and their two wildtypes (US06 and ML13). The superscript letters represent the groups according to Duncan's test.

### 3.2.10. Seed Yield

The results revealed that the studied sesame genotypes presented extensive genetic variability due to the significant differences between the ML genotypes and the US genotypes ($p < 0.001$) and between the mutants and their respective wild-type parents ($p < 0.05$) (Table 4). Seed yield had high PCV and GCV (196.19 and 185.21, respectively), coupled with high $H^2$ b.s (89.13%) and high GA (30.37). The seed yield showed a wide variation ranging from 516 to 1691 kg·ha$^{-1}$, with an overall average value of 1052 kg·ha$^{-1}$ (Table 5). The two mutants 'ML2-5' and ML2-10' were the least productive with a seed yield of 516 and 707 kg·ha$^{-1}$, respectively (Figure 10). However, the mutants 'US2-1' and 'US1-2' produced the highest seed yield of 1691 and 1674 kg·ha$^{-1}$, respectively, which were statistically superior to the parental cultivars (ML13 and US06). These two mutants were 50% more productive than the Moroccan cultivar 'ML13' and 32% more productive than the foreign cultivar 'US06'. In addition to their highest NCP, these two mutants (US2-1 and US1-2) are the most productive in terms of seed yield, compared to all sesame genotypes reported in the literature so far. These performances are higher than those of mutants reported by Bhuiyan et al. [38] (1500 kg·ha$^{-1}$, $H^2$ b.s of 61.5%, and GA of 69.5) and mutants obtained by Laghari et al. [25] (1507.50 kg·ha$^{-1}$ with a $H^2$ b.s of 78.25 and a GA of 1.574). Our data suggest a relevant and significant effect of mutagenesis on the improvement of seed yield in sesame. A positive effect of mutagenesis on sesame seed yield has also been reported in previous studies [25,38].

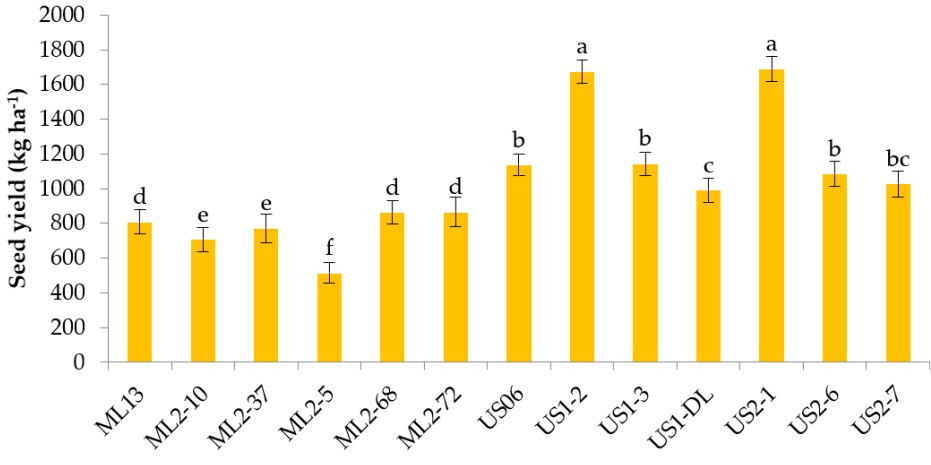

**Figure 10.** Average seed yield (kg·ha$^{-1}$) in 11 sesame mutant lines and their two wildtypes (US06 and ML13). The superscript letters represent the groups according to Duncan's test.

### 3.3. Association among the Investigated Traits

The results of the correlation matrix are summarized in Table 6. There are many strong and significant correlations among the traits studied. The most relevant positive association existed between seed yield and NCP ($r = 0.892$, $p < 0.001$). Additionally, seed yield is significantly and negatively correlated with NDM ($r = -0.628$, $p < 0.001$). This latter was found to be strongly and positively associated with HFC ($r = 0.757$, $p < 0.001$), NDF ($r = 0.786$, $p < 0.001$), and NBP ($r = 0.627$, $p < 0.001$) and negatively correlated with NSC ($r = -0.568$, $p < 0.01$). This demonstrated that the genotypes with most elevated number of capsules per plant and earliest to maturity are the most productive. This is the case of the two highest seed-yielding mutants, 'US1-2' and 'US2-1', having also exhibited the highest NCP (Figure 7). Furthermore, early maturity sesame genotypes (US2-6 and US2-1) appeared to be characterized by reduced HFC, early flowering, low branching, and reduced number of seeds per capsule.

**Table 6.** Pearson's correlation coefficients among the traits investigated.

|            | NDF         | NDM        | HFC         | PH      | FZL       | NBP       | NCP       | NSC      | TSW   | Seed Yield |
|------------|-------------|------------|-------------|---------|-----------|-----------|-----------|----------|-------|------------|
| NDF        | 1           |            |             |         |           |           |           |          |       |            |
| NDM        | 0.786 ***   | 1          |             |         |           |           |           |          |       |            |
| HFC        | 0.861 ***   | 0.757 ***  | 1           |         |           |           |           |          |       |            |
| PH         | 0.328       | 0.380      | 0.451 *     | 1       |           |           |           |          |       |            |
| FZL        | −0.592 **   | −0.470 *   | −0.624 ***  | 0.371   | 1         |           |           |          |       |            |
| NBP        | 0.744 ***   | 0.627 ***  | 0.834 ***   | 0.377   | −0.552 ** | 1         |           |          |       |            |
| NCP        | −0.342      | −0.282     | −0.346      | −0.084  | 0.333     | 0.728 *** | 1         |          |       |            |
| NSC        | −0.614 ***  | −0.568 **  | −0.656 ***  | −0.165  | 0.459 *   | −0.743*** | 0.495 **  | 1        |       |            |
| TSW        | 0.348       | 0.304      | 0.243       | 0.115   | −0.238    | 0.068     | −0.239    | 0.195    | 1     |            |
| Seed yield | −0.410 *    | −0.628 *** | −0.410 *    | 0.178   | 0.353     | −0.316    | 0.892 *** | 0.508 ** | 0.048 | 1          |

NDF, number of days to flowering; NDM, number of days to maturity; HFC, height of the first capsule; PH, plant height; FZL, fruiting zone length; NBP, number of primary branches per plant; NCP, number of capsules per plant; NSC, number of seeds per capsule and TSW, 1000-seed weight. *, ** and ***: correlation is significant at 5%, 1%, and 0.1%, respectively (Correlations are based on 13 observations).

### 3.4. Genetic Relatednes Based on Phenotypical Clustering

Thirteen genotypes were grouped by ascending hierarchical clustering into three (A, B, and C) groups according to Euclidean distance (Figure 11). Group A includes two genotypes, 'US2-6' and 'US2-7' which are characterized by early flowering and maturity, long FZL, low branching, medium NCP, NSC, TSW, and seed yield/ha, and are short in terms of HFC and PH. Group B is subdivided into two subgroups (B1 and B2). This cluster contained the genotypes 'ML2-5', 'ML13', ML2-10', 'US1-DL', 'ML2-68', 'ML-72', and 'ML2-37'. Generally, genotypes of group B were characterized by late flowering and maturity, medium FZL, high branching, low NCP and NSC, low TSW and seed yield/ha, and high HFC and PH. Group C includes four genotypes, namely, 'US2-1', 'US1-2', 'US06', and 'US1-3' bifurcated into two subgroups (C1 and C2). These accessions were characterized by early flowering and maturity, long FZL, low branching, very high NCP, high NSC, TSW, and seed yield. Genotypes of this group had a short HFC and a medium PH. The Mutants 'US2-1' and 'US1-2' of the subgroup (C2) are genetically close, most likely because they both expressed early flowering and early maturity, medium HFC and PH, high FZL, low branching, very high NCP, high NSC, medium TSW, and high seed yield. Overall, genotypes exhibited similarities that align with their agromorphological resemblance and origin, as well (Morocco and Mexico).

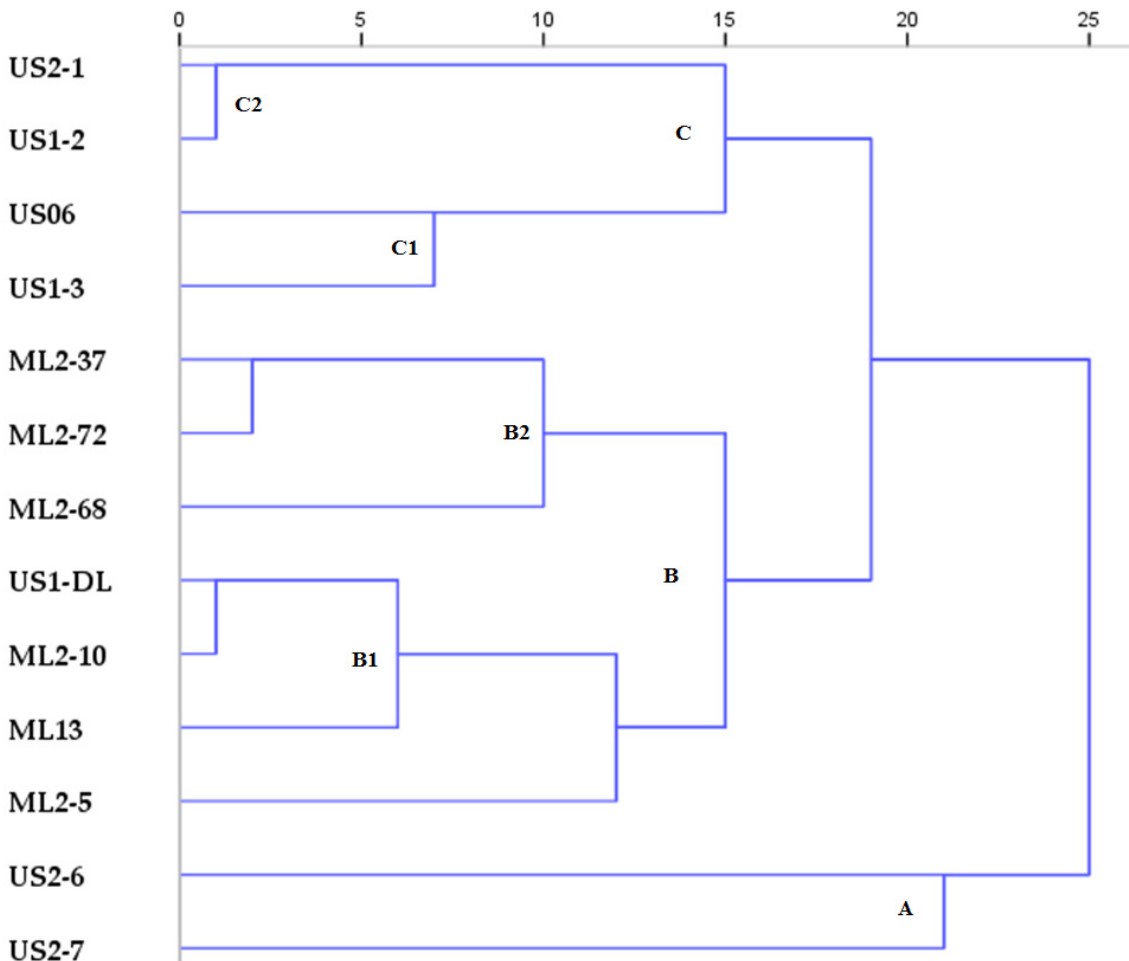

**Figure 11.** UPGMA-based dendrogram depicting genetic relationships among 13 sesame genotypes based on Euclidean dissimilarity estimates for ten agromorphological traits.

### 4. Conclusions

This study aimed to characterize and compare 11 sesame mutant lines M4 along with their two wild-type cultivars across two different environments. Wide genetic variability was observed among the mutants studied and between mutants and their respective parents, indicating a genetic gain achievement for most of the traits, as a result of mutagenesis breeding. The mutants 'US2-6' and 'US2-1' matured early, about 24 days before the Moroccan check cultivar (ML13) and one week before the Mexican cultivar (US06). Moreover, both the mutants 'US2-1' and 'US1-2' had the highest seed yield ever reported, being 50% more productive than the Moroccan cultivar 'ML13' and 32% more productive than the Mexican cultivar 'US06'. Compared to these check cultivars, the mutants 'US1-3' and 'US2-6' were earlier at flowering (about 12 days), suggesting their use in dry regions to reduce water requirements. Moreover, the high levels of estimated broad-sense heritability, obtained on most traits, make their genetic improvement easy through traditional breeding. Therefore, these promising mutants (US2-1, US1-2, US1-3, and US2-6) are worth considering in crossbreeding programs to develop competitive sesame cultivars to meet the increasing demand for sesame oil and seeds in the actual context of climate change. Finally, one could conclude that EMS-mutagenesis used allowed inducing a novel genetic variability in sesame for almost all the desirable traits and obtaining highly performing mutant lines exhibiting a remarkable genetic progress, compared to the wild-type cultivars. Mutagenesis is, thus, a biotechnological tool that could be applied successfully for the genetic improvement of sesame, mainly where the existing genetic variability is too limited.

**Author Contributions:** Conceptualization, investigation, data curation, writing—original draft, M.K.; supervision, conceptualization, validation, and writing—review and editing and project administration, A.N.; writing—review and editing, O.C.; investigation, M.E.F. and S.F.; supervision, conceptualization, writing—review and editing, H.H. All authors have read and agreed to the published version of the manuscript.

**Funding:** This research received no external funding.

**Institutional Review Board Statement:** Not applicable.

**Informed Consent Statement:** Not applicable.

**Data Availability Statement:** Not applicable.

**Conflicts of Interest:** The authors declare no conflict of interest.

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
