# Peer review of "Assessment of Novel Genetic Diversity Induced by Mutagenesis and Estimation of Genetic Parameters in Sesame M4 Mutant Lines"

_2037-0164, doi:10.3390/ijpb13040052_

Round 1
Reviewer 1 Report
This is an interesting paper showing how mutagenesis can effectively be used in widening the genetic diversity in sesame. The paper is very well written and presented. The introduction is good, but could be improved by including some more information about the the crop and its varietal diversity and on its geographical distribution both in Morocco and elsewhere. The presentation of the results could also be improved by having some graphical representation of the data rather solely by tables. I think the results is well discussed and good conclusion.
Otherwise I recommend publication of the a paper with minor revisions, taking my comments above into account.
Author Response
First of all, we would like to thank you and the other Reviewer for your reviews and valuable and useful comments. All the comments were taken into consideration for the correction and improvement of our manuscript. Please note that all modifications, amendments and editing in the revised manuscript are highlighted in yellow. Hereafter are given our responses to the specific comments.
Reviewer 1
Comment 1: The introduction is good, but could be improved by including some more information about the crop and its varietal diversity and on its geographical distribution both in Morocco and elsewhere.
- Response: As highlighted in manuscript, we have provided additional information on the crop (Lines 40-43), on its geographical distribution in Morocco and over the world (Lines 47-49 and lines 52-54). The sesame varietal diversity in Morocco was already mentioned in the initial version of our manuscript (Lines 56-58 of the revised manuscript). However, new information was provided with regard to the genetic diversity reported in most of the producing countries of this crop (Lines 59-61).
Comment 2: The presentation of the results could also be improved by having some graphical representation of the data rather solely by tables.
- Response: As suggested, average values of the ten traits studied in the 13 genotypes are now presented in histogram graphics (Figures 1-10).
Reviewer 2 Report
There is a lack of information on the method of propagation of seeds of mutant lines (isolation, open flowering)
It is desirable to provide data on the correlation analysis of yields with other traits.
It is not clear what caused the significant increase in seed yield in mutant lines US2-1 and US2-2.
Author Response
First of all, we would like to thank you and the other Reviewer for your valuable reviews and comments. All the comments were taken into consideration for the correction and improvement of our manuscript. Please note that all modifications, amendments and editing in the revised manuscript are highlighted in yellow. Hereafter are given our responses to your specific comments.
Reviewer 2
Comment 1: There is a lack of information on the method of propagation of seeds of mutant lines (isolation, open flowering).
- Response: The missing information has been provided through lines 105-108 of the revised manuscript. Seeds of the mutant lines are propagated in isolation to keep the authenticity of the genetic material as cross pollination, even with a low rate, can occur in sesame. In fact, the selected mutant plants are bagged before flowering to ensure self-fertilization by avoiding any contamination by allopollen.
Comment 2: It is desirable to provide data on the correlation analysis of yield with other traits.
- Response: As suggested, we added Pearson’s correlation analysis to reveal the relationships between yield and other traits (Please see lines 176-177 and the added subsection 3. Association among the investigated traits).
Comment 3: It is not clear what caused the significant increase in seed yield in mutant lines US1-2 and US2-1
- Response: From the results of the Pearson’s correlation and as shown in the figure 7, the trait most contributing to the high yield in the mutants ‘US1-2’ and ‘US2-1’ is their highest number of capsules per plant, compared to the other mutants and wild-types. Please see our discussion in the lines 456-457, 461-464.
Reviewer 3 Report
This manuscript studies an important crop species for human nutrition like sesame. A common problem for sesame in most countries, is the limited genetic diversity. So, this manuscript could have important contribution in the creation of novel variability in sesame
Abstract is well written. It is referred the aim, the experimental design, the main conclusion of the experiment
Introduction is good. The aim of the experiment is presented very clear at the end of the introduction. I proposed to the authors to add a few lines about the importance of sesame in human nutrition, in the first sentence (lines 38-50)
Material and Methods, Results, Discussion are well written.
Conclusions need some changes. A senond sentences should be added after line 455, in which authors should refer the general conclusion of their work e.g. this method of mutagenesis help in the creation of novel variability in sesame for almost all agronomic important characteristics …….. could be applies for the improvement of sesame ……..

Author Response
Responses to reviewer’s comments
First of all, we would like to thank the three Reviewers for their valuable reviews and comments. All the comments were taken into consideration for the correction and improvement of our manuscript. Please note that all modifications, amendments and editing in the revised manuscript are highlighted in yellow. Hereafter are given our responses to the specific comments.
Reviewer 3
Comment 1: I proposed to the authors to add a few lines about the importance of sesame in human nutrition, in the first sentence (lines 38-50).
- Response: Additional information on the nutritional importance of sesame seeds and oil was provided in the revised manuscript (Lines 41-47).
Comment 2: Conclusions need some changes. A second sentence should be added after line 455, in which authors should refer the general conclusion of their work
- Response: The conclusion section was amended as suggested (Lines 522-527).